# Effects of Dietary Supplementation of Peanut Skin Proanthocyanidins on Growth Performance and Lipid Metabolism of the Juvenile American Eel (*Anguilla rostrata*)

**DOI:** 10.3390/ani12182375

**Published:** 2022-09-12

**Authors:** Yue Wang, Xue-Hao Chen, Xin-Yi Wu, Guo-He Cai, Shao-Wei Zhai

**Affiliations:** Engineering Research Center of Modern Industry Technology for Eel, Ministry of Education of PRC, Fisheries College of Jimei University, Xiamen 361021, China

**Keywords:** *Anguilla rostrate*, peanut skin, proanthocyanidins, growth performance, serum lipid, lipid metabolic enzyme, lipidomics, autophagy, sphingolipid metabolism, glycerophospholipid metabolism, linoleic acid metabolism

## Abstract

**Simple Summary:**

Proanthocyanidins, mainly extracted from grape seed, receive considerable attention due to their biological activity in the health of aquatic animals. High costs limit the application of this functional feed additive in aquaculture. A new resource of proanthocyanidins is urgent to explore for sustainable aquaculture. In the present study, we assessed and proved the potential of peanut skin proanthocyanidins as a new feed additive in American eel, expressed as promoted growth performance and regulated lipid metabolism involving decreased lipid levels in whole fish and serum, altering the activities or levels of lipid metabolic enzymes and certain lipid metabolites and lipid metabolic pathways in the liver of this fish species.

**Abstract:**

As a functional feed additive, grape seed proanthocyanidin extract has received a lot of attention due to its biological activity in the health of aquatic animals, but its high cost limits the application of this feed additive in the diet of many fish species. It is thus urgent to develop a new resource of proanthocyanidin extract. We aimed to investigate the effects of dietary supplementation with peanut skin proanthocyanidins (PSPc) on growth parameters and lipid metabolism of juvenile American eel (*Anguilla rostrata*). Four hundred and fifty juvenile eels were randomly divided into five groups fed diets with five PSPc supplementation levels. The trial lasted for 8 weeks. Dietary PSPc supplementation significantly improved weight gain and feed utilization, and the best growth performance was found in the group fed with 900 mg/kg PSPc. PSPc supplementation significantly affected the crude protein level of whole fish and serum lipid parameters, and the best lipid-lowering effect was found in the fish fed with 900 mg/kg PSPc. Dietary PSPc supplementation increased lipolytic enzyme activities and decrease lipid synthase levels in the liver. The lipid metabolites affected by 900 mg/kg PSPc in the liver were mainly upregulated phosphatidylethanolamine in autophagy, downregulated ceramides in sphingolipid metabolism, upregulated phosphatidylcholine and phosphatidylethanolamine, downregulated 2-lysophosphatidylcholine in glycerophospholipid metabolism, and upregulated phosphatidylcholine in linoleic acid metabolism. In conclusion, an appropriate level of PSPc might effectively improve growth performance and regulate the lipid metabolism of the juvenile American eel, and 900 mg/kg PSPc is recommended in the diet of this fish species.

## 1. Introduction

Proanthocyanidins (PACs) are oligomers or polymers from the condensation of two or more flavan-3-ol units, and they may differ from each other in terms of the number and position of hydroxyl groups linked to the aromatic rings, the stereochemistry of flavonol heterocycle, and the type of linkage among the different units [1]. Proanthocyanidins are primarily known for their strong antioxidant activity, which is due to their effective hydrogen donation, as well as the effective delocalization of an unpaired electron. Based on strong antioxidant potential, proanthocyanidins are attracting considerable interest in the nutraceutical field due to their potential human health benefits, including their systemic hypoglycemic and lipid-lowering effects and their local anti-inflammatory actions on the intestinal epithelium [1,2]. Proanthocyanidins are also applied as natural antioxidants in food products to extend their shelf life by replacing synthetic antioxidants, and in cosmetic products for anti-aging reasons [2]. Proanthocyanidins are usually extracted from grape seed or pine bark. They are also present in red wines, hops, and various flowers, leaves, fruits, berries, nuts, and beans [1,2]. There are high concentrations of proanthocyanidins in skins, barks, seeds, or other waste products of the agribusiness industry. Therefore, the use of this waste product is important from a circular economy point of view.

With growing interest in the utilization of phytogenic feed additives as an eco-friendly measure to ensure the sustainability of aquaculture, proanthocyanidins have been widely reported as a functional feed additive to exert growth promotion effects and improve the health status of some fish species, including the American eel (*Anguilla rostrata*), common carp (*Cyprinus carpio*), hybrid sturgeon (*Acipenser baeri* Brandt ♀ × *A. schrenckii* Brandt ♂), rainbow trout (*Oncorhynchus mykiss*), and tilapia (*Oreochromis niloticus*) [3,4,5,6,7,8,9,10]. The hypolipidemic effect of proanthocyanidins has attracted much attention for its potential to improve the health of humans [1] and some fish species, including the American eel, grass carp (*Ctenopharyngodon idella*), and tilapia [3,4,10,11]. At present, the proanthocyanidins reported in aquatic animals are mainly from grape seed proanthocyanidins (GSP). Other sources of proanthocyanidins are seldom reported in the aquaculture field.

Peanut skin is another source of biologically active polyphenols that contain up to 17% proanthocyanidins [12]. They are usually discarded as waste or used as feedstuff for animal diets [13]. Peanut skin proanthocyanidins (PSPc) have been given significant attention due to their similar biological activity and lower cost in comparison to proanthocyanidins from grape seed, but far less is known regarding the application of PSPc supplemented in the diets of aquatic animals.

Eels, reputed as “ginseng in water”, are one of the most common fish species cultured around the world, and they play an important role in international trade [14]. The American eel has become the main cultured eel species in Asia since European eels (*Anguilla anguilla*) have been listed as extremely endangered species [14,15]. The purpose of the present study was to investigate the effects of dietary supplementation of PSPc on the growth performance and lipid metabolism of juvenile American eels.

## 2. Materials and Methods

### 2.1. Feeding Trial

The trial fish was the juvenile American eel, purchased from Fujian Jinjiangzhiman Aquatic Technology Co., Ltd., Zhangzhou, China. Before the formal trial, all fish were acclimatized in two PVC tanks. The two tanks have a water volume of about 1200 L and are supplied with 5 L/min of degassed and dechlorinated municipal water. The trial fish were fed a commercial powder feed to apparent satiation twice daily (6:00 a.m.; 6:00 p.m.). The commercial diet was manufactured by Fuzhou Xinruiyi Industrial Co., Ltd., Fuzhou, China. The diet in powder form was mixed with water 1.1 times the dry diet weight to form a dough, then placed on a feeding table for the eels to consume. After feeding for 30 min, the uneaten feed was taken out with a net, then dried to record the feed consumption of each tank daily. The proximate composition of the commercial diet was 46.14% crude protein, 5.17% crude fat, 14.99% ash, and 5.44% moisture. During cultivation, the water was monitored daily, and the quality parameters were maintained at 24–26 °C, pH 7.1–7.6, dissolved oxygen ≥ 8.0 mg/L, and ammonia nitrogen levels ≤ 0.25 mg/L.

After the adaptation period of 4 weeks, 450 trial fish (10.50 ± 0.03 g/fish) were randomly divided into five groups, which were the PSPc0 group, PSPc300 group, PSPc600 group, PSPc900 group, and PSPc1200 group. These five groups were fed a basal diet, and the basal diet with PSPc supplementation levels of 300 mg/kg, 600 mg/kg, 900 mg/kg, and 1200 mg/kg. Each group had three replicates, with 30 fish in each replicate. The trial period was 8 weeks. The basal diet was the same as the commercial diet in the adaption period. PSPc (provided by Shandong Jinsheng Biological Technology Co., Ltd., Linyi, China, the powder product with a content of 95%) was mixed with the basal diet. The juvenile American eels were cultured in 15 circular PVC tanks (400 L of water in each tank) with a water recirculation system. During the formal trial period, the water quality and fish management were the same as those in the adaption period.

### 2.2. Sample Collection

At the end of the feeding trial, all the fish were starved for 12 h before sampling. Fifteen fish per tank were randomly collected to anesthetize with 100 mg/L eugenol. Five intact fish were randomly captured and stored at −20 °C for proximate composition analysis. Blood was taken from the caudal vein of 10 fish in each tank and mixed to analyze serum lipid parameters according to Liu et al. [16]. The liver samples were aseptically removed and frozen immediately in liquid nitrogen, and then stored at −80 °C for subsequent analysis.

### 2.3. Calculations of Growth Performance Parameters

After 12 h starvation at the end of the feed trial, all the fish in each tank were weighed and counted to calculate the parameters related to growth performance by the following equations, which were used in the previous studies [3,4,5,6,7,8,9,10]:(1)$$\mathrm{Initial}\;\mathrm{body}\;\mathrm{weight}\;(\mathrm{IBW},\;g/\mathrm{fish})=\frac{\mathrm{Initial}\;\mathrm{weight}\;\mathrm{of}\;\mathrm{fish}\;\left(g\right)}{\mathrm{Initial}\;\mathrm{fish}\;\mathrm{number}}$$
(2)$$\mathrm{Final}\;\mathrm{body}\;\mathrm{weight}\;(\mathrm{FBW},\;g/\mathrm{fish})=\frac{\mathrm{Final}\;\mathrm{weight}\;\mathrm{of}\;\mathrm{fish}\;\left(g\right)}{\mathrm{Final}\;\mathrm{fish}\;\mathrm{number}\;}$$
(3)$$\mathrm{Weight}\;\mathrm{gain}\;\mathrm{rate}\;(\mathrm{WGR},\;\%)=100\;\times \;\frac{\mathrm{Final}\;\mathrm{fish}\;\mathrm{weight}\;\left(g\right)-\;\mathrm{Initial}\;\mathrm{fish}\;\mathrm{weight}\;\left(g\right)}{\mathrm{Initial}\;\mathrm{fish}\;\mathrm{weight}\left(g\right)}$$
(4)$$\mathrm{Feed}\;\mathrm{efficiency}\;(\mathrm{FE},\;\%)=100\;\times \;\frac{\mathrm{Final}\;\mathrm{fish}\;\mathrm{weight}\;\left(g\right)-\;\mathrm{Initial}\;\mathrm{fish}\;\mathrm{weight}\left(g\right)}{\mathrm{Feed}\;\mathrm{intake}\;\left(g\right)}$$
(5)$$\mathrm{Feeding}\;\mathrm{rate}\;(\mathrm{FR},\;\%)=100\;\times \;\frac{\mathrm{Feed}\;\mathrm{intake}\;\left(g\right)}{\mathrm{Trial}\;\mathrm{days}\;\left(d\right)\times \left(\mathrm{Initial}\;\mathrm{fish}\;\mathrm{weight}\;\left(g\right)+\;\mathrm{Final}\;\mathrm{fish}\;\mathrm{weight}\;\left(g\right)\right)\;÷\;2}$$
(6)$$\mathrm{Survival}\;\mathrm{rate}\;(\mathrm{SR},\;\%)=100\;\times \;\frac{\mathrm{Final}\;\mathrm{fish}\;\mathrm{number}}{\mathrm{Initial}\;\mathrm{fish}\;\mathrm{number}}$$

The weight and number of fish in the equations were the values of each tank.

### 2.4. Proximate Composition Analysis

The crude protein content (using the Kjeldahl apparatus, nitrogen × 6.25), crude lipid content (petroleum ether extraction using the Soxhlet principle), moisture content (drying samples at 105 °C until a constant weight is achieved), and ash content (oven incineration at 550 °C for 5 h) of the whole fish were determined according to the methods of AOAC [17].

### 2.5. Measurement of Serum Lipids and Lipid Metabolic Enzymes in the Liver

Commercial kits (manufactured by Nanjing Jiancheng Bioengineering Institute, Nanjing, China) were used to measure the serum lipid levels, including total cholesterol (TC), triglycerides (TG), high-density lipoprotein cholesterol (HDL-C), and low-density lipoprotein cholesterol (LDL-C).

The liver sample was homogenized according to Liu et al. [16]. The activities of lipoprotein lipase (LPL), hepatic lipase (HL), and total lipase (TL) and the levels of fatty acid synthetase (FAS), and acetyl-CoA carboxylase (ACC) in the liver were determined by commercial kits according to the instructions provided by Nanjing Jiancheng Biological Engineering Institute (Nanjing, China).

The experimental protocols for measuring the above parameters were the same as in the previous studies [3,18].

### 2.6. Lipidomics Profiling in the Liver

The treatment and preparation of the liver sample for lipidomic analysis were the same as described by Ma et al. [18]. First, 100 mg of each liver sample was transferred into 2 mL centrifuge tubes. Then, we added 750 μL of chloroform–methanol mixed solution (2:1) (pre-cooled at −20 °C) and vortexed for 30 s. Next, we added 2 steel balls, put the samples into the Xinzhi high-flux tissue grinder (SCIENTZ-48, Ningbo, China), and ground them for 90 s at 55 Hz. The mixture was put on ice for 40 min. We added 190 μL H_2_O, vortexed for 30 s, and put the mixture on ice for 10 min. We then centrifuged the mixture at 12,000 rpm for 5 min at room temperature and transferred 300 μL of the lower layer fluid into a new centrifuge tube. Then, we added 500 μL of the chloroform–methanol mixed solution (2:1) (pre-cooled at −20 °C), vortexed for 30 s, and centrifuged at 12,000 rpm for 5 min at room temperature. We transferred 400 μL of the lower layer fluid into the same centrifuge tube as above. Samples were concentrated to dryness in a vacuum; then, dissolved samples with 200 μL of isopropanol and the supernatant were filtered through a 0.22 µm membrane to obtain the prepared samples for LC-MS. To avoid instrumentation error, 20 µL from each sample to the quality control (QC) samples were prepared to evaluate stability and reliability during the experiment. Dynamic exclusion was implemented to remove some unnecessary information in MS/MS spectra.

The lipidomic analysis was performed by liquid chromatography–mass spectrometry (LC-MS) in Suzhou Panomix Biotechnology Co., Ltd. (Suzhou, China). The liquid chromatography (UltiMate 3000, Thermo Fisher Scientific, San Diego, CA, USA) was coupled to a mass spectrometer (Q Exactive Focus, Thermo Fisher Scientific, San Diego, CA, USA). The parameters of chromatographic columns and separation conditions, mass spectrometry scan mode and collision mode, methods of data processing, and statistical analysis in LC-MS analysis were the same as described in the study by Li et al. [19]. Chromatographic separation was accomplished in a Thermo Ultimate 3000 system equipped with an ACQUITY UPLC^®^ BEH C18 (100 × 2.1 mm, 1.7 µm, Waters, Milford, MA, USA) column maintained at 50 °C. The temperature of the autosampler was 8 °C. Gradient elution of analytes was carried out with acetonitrile:water = 60:40 (0.1% formic acid +10 mM ammonium formate) (C) and isopropanol:acetonitrile = 90:10 (0.1% formic acid +10 mM ammonium formate) (D) at a flow rate of 0.25 mL/min. The injection of 2 μL of each sample was carried out after equilibration. An increasing linear gradient of solvent C (*v*/*v*) was used as follows: 0–5 min, 70–57% C; 5–5.1 min, 57–50% C; 5.1–14 min, 50–30% C; 14–14.1 min, 30% C; 14.1–21 min, 30%–1% C; 21–24 min, 1% C; 24–24.1 min, 1%–70% C; 24.1–28 min, 70% C. The ESI-MSn experiments were executed on the Thermo Q Exactive Focus mass spectrometer with spray voltage of 3.5 kV and −2.5 kV in positive and negative modes, respectively. Sheath gas and auxiliary gas were set at 30 and 10 arbitrary units, respectively. The capillary temperature was 325 °C. The Orbitrap analyzer scanned over a mass range of m/z 150–2000 for the full scan at a mass resolution of 35,000. Data-dependent acquisition (DDA) MS/MS experiments were performed with an HCD scan. The normalized collision energy was 30 eV. Dynamic exclusion was implemented to remove some unnecessary information in MS/MS spectra.

### 2.7. Statistical Analysis

The data of growth performance, serum lipids, and lipid metabolic enzymes in the liver from different PSPc groups were subjected to Duncan’s multiple comparisons in a one-way ANOVA model to estimate the statistical significance (*p* < 0.05) by SPSS 23.0 statistical software (SPSS Inc., Chicago, IL, USA). All the data of the above parameters were presented as means ± standard deviation. The results, expressed as percentages, were subjected to square arcsine transformation before statistical analysis. Orthogonal partial least squares discriminant analysis (OPLS-DA) was used to assess the annotated raw data based on the Lipid Structure Database (LMSD; http://www.lipidmaps.org/ (accessed on 12 March 2021)). The statistical significance of lipid types was confirmed by variable importance in projection (VIP) > 3.0, fold change (FC) > 2.00 or < 0.67, and *p* < 0.01. The differential lipid metabolites and enriched metabolic pathways were searched using the Kyoto Encyclopedia of Genes and Genomes (KEGG) pathway database (https://www.kegg.jp/kegg/ (accessed on 20 January 2022)).

## 3. Results

### 3.1. Growth Performance and Proximate Composition of Whole Fish

The growth performance of juvenile American eels in different PSPc groups is shown in Table 1. Compared with the PSPc0 group, the FBW, WGR, and FE (except for the PSPc300 group) of the PSPc supplementation groups were significantly increased (*p* < 0.05), and the FR only in the PSPc300 group and PSPc900 group were significantly improved (*p* < 0.05). There was no significant difference in SR among all PSPc groups (*p* > 0.05).

The proximate composition of whole fish of juvenile American eel in different PSPc groups is given in Table 2. Compared with the PSPc0 group, crude lipid levels of the PSPc supplementation groups (except for the PSPc300 group) were significantly decreased (*p* < 0.05). There was no significant difference in the levels of crude protein, ash, and moisture among all the PSPc groups (*p* > 0.05).

### 3.2. Serum Lipid Parameters

The serum lipid parameters of juvenile American eel in different PSPc groups are presented in Table 3. Compared with the PSPc0 group, the levels of TG and LDL-C were significantly decreased in the PSPc supplementation groups (*p* < 0.05), while HDL-C levels of the PSPc supplementation groups were significantly increased (*p* < 0.05), and the TC level was significantly decreased only in PSPc900 group (*p* < 0.05).

### 3.3. Lipid Metabolic Enzymes in the Liver

The lipid metabolic enzyme activities or levels in the liver of juvenile American eels in different PSPc groups are given in Table 4. Compared with the PSPc0 group, higher activities of LPL, HL, and TL were observed in the PSPc supplementation groups (*p* < 0.05), and lower levels of FAS and ACC were shown in the PSPc supplementation groups (*p* < 0.05).

### 3.4. Lipidomics Profiling in the Liver

The score plots and validation plots of OPLS-DA for lipid profiles in the liver of juvenile American eels in the PSPc0 and PSPc900 groups are shown in Figure 1A,B, respectively. The higher scores and validation indexes of OPLS-DA indicate that there might be a robust model with a low risk of overfitting and reliability. The samples of the two groups were well separated, implying that the OPLS-DA model could be utilized to identify the differences between the two groups.

There were 204 lipids belonging to 23 subclasses in the livers of the juvenile American eels in the PSPc0 group and PSPc900 group. Ultimately, 20 kinds of lipids (VIP > 3.0, *p* < 0.01) were selected as significantly different lipids between the PSPc0 group and the PSPc900 group. As shown in Table 5, the 20 lipids were primarily methyl phosphatidylcholine (MePC), phosphatidylcholine (PC), sphingomyelin (SM), phosphatidylethanolamine (PE), phosphatidylethanol (PEt), lysophosphatidylethanol (LPEt), and phosphatidylmethanol (PMe). Z-score analysis was performed on 20 significantly different lipids (Figure 2). In Figure 2, the x-coordinate is the Z-score value and the y-coordinate is the different lipids. Each circle or triangle represents a sample of the PSPc0 group or PSPc900 group. The Z-score value is converted based on the relative lipid content with PSPc0 as the reference data set.

### 3.5. Lipid Metabolism Pathways and Related Lipid Metabolites

The differential metabolic pathways by the functional enrichment analysis on 204 lipid metabolites in the PSPc0 group and PSPc900 group are shown in Table 6. Autophagy, sphingolipid metabolism, glycerophospholipid metabolism, and linoleic acid metabolism were the major metabolic pathways (Figure 3). PE was upregulated in autophagy, ceramides (Cer) were downregulated in sphingolipid metabolism, PC and PE were upregulated and 2-lysophosphatidylcholine (2-LPC) was downregulated in glycerophospholipid metabolism, and PC was upregulated in linoleic acid metabolism.

## 4. Discussion

### 4.1. Effects of Dietary PSPc Supplementation on Growth Performance and Proximate Composition of Whole Fish of the Juvenile American Eel

In the present study, appropriate levels of dietary PSPc supplementation increased the WGR, SGR, FE, and FR of the juvenile American eel. Our results indicate that dietary PSPc supplementation might have a positive impact on growth performance. Wang et al. [15] demonstrated that dietary 400 mg/kg GSP could improve the WGR of the American eel, which is in line with the results of the present study. Similarly, the WGR and FE of tilapia were improved by 200–800 mg/kg dietary GSP [4]. In the studies of rainbow trout, 1000 mg/kg grape seed extract supplemented in the diet improved WGR, SGR, and FE [6], while 1000 mg/kg grape seed oil increased WGR and SGR [5]. Mehrinakhi et al. [8] found that 10–30 g/kg dietary grape seed extract improved the WGR and FE of common carp. Furthermore, dietary GSP supplementation could ameliorate the growth retardation of the American eel exposed to dietary histamine stress [7] and pearl gentian grouper (*Epinephelus fuscoguttatus* female × *Epinephelus lanceolatus* male) and tilapia exposed to dietary cadmium stress [20,21]. The possible reasons for the growth-promoting effect of PSPc might be the same as the GSP—PSPc could be absorbed in the anterior intestine and quickly metabolized to induce the synthesis of proteins associated with cytoskeletal function, resulting in an increase in the small intestine absorptive surface, and further promote the absorption of nutrients [22]. The absorbed PSPc could improve the intestinal digestive enzyme activities and positively regulate the intestinal microbiota to maintain intestinal health and eventually improve growth performance [10]. In the present trial, the best growth performance of the American eel was found in the group fed 900 mg/kg PSPc, which was higher than the optimal level of GSP at 400 mg/kg. The difference in optimal level between PSPc and GSP might be due to the different structure and composition of proanthocyanidins. The catechin and epicatechin are the basic structural units of proanthocyanidins. The quantity and degree of polymerization of catechin and epicatechin are different between PSPc and GSP [23]. The proanthocyanidins in PSPc belong to the A-type, and the proanthocyanidins in GSP belong to the B-type. The main difference between A-type and B-type proanthocyanidins is the interflavanic linkages, and the A-type at least has one double linkage consisting of a C-C bond and an additional ether bond.

Lower levels of crude lipids in whole fish were found in juvenile American eels in PSPc600, PSPc900, and PSPc1200 groups. This result was similar to the study of tilapia fed 200–800 mg/kg GSP [4]. The decreasing lipid levels in whole fish indicated that the PSPc might have a hypolipidemic effect in juvenile American eel. However, 200 mg/kg dietary GSP could increase the fillet protein content of common carp, and 500 mg/kg grape seed extract supplemented in the diet enhanced the protein level of the whole body of rainbow trout [6,9]. The difference in the above results might be due to fish species, resources and supplementation levels of proanthocyanidins, and dietary nutrient levels.

### 4.2. Effects of Dietary PSPc Supplementation on Serum Lipids and Lipid Metabolic Enzymes of the Juvenile American Eel

The levels of TC, TG, LDL-C, and HDL-C in serum are relevant to lipid metabolism; excessive TC and TG in the blood are considered to result in metabolic diseases [3]. In the present study, there were lower levels of TC, TG, and LDL-C and higher HDL-C levels in the PSPc-treated groups, and there was no other report about the effect of PSPc on serum lipid in fish. The results of the present trial were consistent with the study of American eels fed 400 mg/kg GSP [3]. Similar results were also found in common carp and tilapia fed 200 mg/kg or 400 kg/kg GSP; the levels of TC and TG in the serum of those two fish species were found to be decreased significantly [4,9]. However, GSP could not reduce the TG levels in the serum of rainbow trout [24]. The lipid-reduced mechanisms in serum of PSPc might be related to inhibiting cholesterol absorption and synthesis, stimulating LDL-C receptor transcription, and downregulating certain genes of TG synthesis, along with upregulating some genes of lipid catabolism [11,25].

The liver is an important organ for fatty acid oxidation [26]. The LPL and HL are parts of lipolytic enzymes in the fish liver and are collectively known as TL. The LPL can hydrolyze TG-rich lipoprotein and produce free fatty acid. The HL can be used as a ligand to promote LDL and chylomicron remnant entry into liver cells [26,27]. The FAS and ACC are related to lipid synthesis in the liver. The FAS is a key enzyme to regulate the de novo biosynthesis of long-chain fatty acids. The ACC is mainly responsible for the first step of de novo fatty acid biosynthesis [28]. In the present study, there were increasing activities of LPL, HL, and TL and decreasing levels of FAS and ACC in the liver of the PSPc-treated groups, and little information was available about the effect of PSPc on lipid metabolism enzymes in the fish liver. However, Lu et al. revealed that GSP might reduce fat accumulation and fatty acid synthesis by upregulating the LPL gene expression and downregulating the gene expression of FAS and ACC in the liver after gavage of grass carp with 250 mg/kg GSP for 3 h [11].

### 4.3. Effects of Dietary PSPc Supplementation on Metabolites and Metabolic Pathways Involved in Lipid Metabolism in the Liver of the Juvenile American Eel

In recent years, lipidomics has been widely applied to reveal the internal changes in organisms through the comprehensive analysis of subtle changes in lipids at the molecular level [29]. The liver is the central organ of lipid metabolism in fish species [30]. In the present study, 900 mg/kg dietary PSPc upregulated the levels of PC and PE, downregulated the levels of Cer and 2-LPC, and mainly affected the pathways of autophagy, sphingolipid metabolism, glycerophospholipid metabolism, and linoleic acid metabolism in the liver.

Autophagy is an intracellular degradation system and is key to maintaining homeostasis in the cells [31]. A previous study reported that the autophagosome could transfer lipid droplets to lysosomes and inhibit lipid accumulation [32]. On the contrary, the inhibition of autophagy triggered the TG increase and lipid droplet accumulation in hepatocytes [33]. The LC3 is an autophagy marker protein that reflects the procession of autophagy. The PE plays a role in autophagy by covalently attaching to modify LC3 and is located on the autophagosome membrane, consequently mediating autophagosome formation and elongation [34,35]. Therefore, the upregulated PE level in the autophagy pathway is likely to induce adipocyte autophagy and reduce lipid accumulation in the liver. Furthermore, a study on zebrafish (*Danio rerio*) showed that polyphenols might induce the autophagy process, reduce lipid droplet formation, and promote lipid droplet breakdown in the liver [35], which was similar to the present study. These results suggest that PSPc might have positive regulatory roles in lipid metabolism through autophagy.

Sphingolipids are the bioactive lipids that regulate the major biological processes and contribute to the arrangement of membrane lipid domains [36], including four subclasses: phosphosphingolipid, glycosphingolipids, sphingoid bases, and Cer. A lipidomic analysis showed that sphingolipid content (especially Cer) might be strongly correlated with the TG level in the liver, and a significant increase in hepatic Cer content was observed in rats fed with a high-fat diet [37]. In addition, ceramide could activate the pro-inflammatory pathways in macrophages and secrete pro-inflammatory cytokines TNF-α and interleukin-1 [36]. The downregulation of Cer in the liver of the PSPc900 group suggested that there might be a reduction in the liver lipid accumulation and inflammation by inhibiting the sphingolipid metabolism.

Glycerophospholipids, such as PC and PE, are major components of biological membranes. PC seems to be the predominant phospholipid class in lipoproteins of fish [38,39]. PE is one of the components of the cell membrane system, and the majority of PE is located in the inner lipid bilayer structure [40]. A recent study showed that PC could increase the expression of LPL and hormone-sensitive lipase and decrease the FAS expression in the liver of tilapia [41]. PE is required for the function and integrity of mitochondrial membranes [34] and is used to help TC metabolism by transforming into PC [42]. LPC is an important signaling molecule for many diseases and could produce pro-inflammatory actions, including the release of chemotactic factors and the production of reactive oxygen species [43,44]. In this study, the glycerophospholipid metabolism pathway was significantly affected by involving the downregulated level of the 2-LPC in the liver of the American eels in the PSP900 group. Similarly, the downregulation of hepatic LPC was also found in black seabream fed a diet with polyphenol supplementation [44]. In general, 900 mg/kg dietary PSPc could modulate the glycerophospholipid metabolism by upregulating levels of PC and PE and downregulating 2-LPC levels in the liver of juvenile American eels, which might be beneficial in decreasing liver lipid accumulation.

As an essential fatty acid for vertebrates, linoleic acid was found to accelerate fatty acid oxidation [45]. It has been reported that the content of PC in rat liver microsomes is positively proportional to the level of linoleic acid [46]. Therefore, the upregulation of PC might have enhanced the anabolism of linoleic acid in the present study, and the increased linoleic acid level could further promote the proliferation and differentiation of fatty cells to exert hypolipidemic effects [45,46]. In addition, linoleic acid could increase the number of LDL receptors and enhance their affinity with LDL to accelerate LDL degradation to reduce the TC level [46], which supports the results of the decreased TC level in the serum of the PSPc900 group.

In addition, PACs from grape seed were reported to interfere with lipid metabolism involved in intestinal absorption of lipids, changes in the gut microbiota to enhance microbial propionate production, and the secretion of chylomicrons and lipoproteins by the intestine and liver of terrestrial animals [1,47]. The mechanism of lipid metabolism regulation by dietary PSPc supplementation should be clarified in this regard in future studies on the American eel.

## 5. Conclusions

The results of the present study indicate that the appropriate level of dietary PSPc supplementation could promote growth performance and regulate lipid metabolism in American eel by reducing levels of whole fish lipid and serum lipid, regulating the activities of lipid metabolism enzymes and certain lipid metabolites involved in autophagy, sphingolipid metabolism, glycerophospholipid metabolism, and linoleic acid metabolism in the liver. A diet supplemented with 900 mg/kg PSPc is recommended for the juvenile American eel.

## Figures and Tables

**Figure 1 animals-12-02375-f001:**
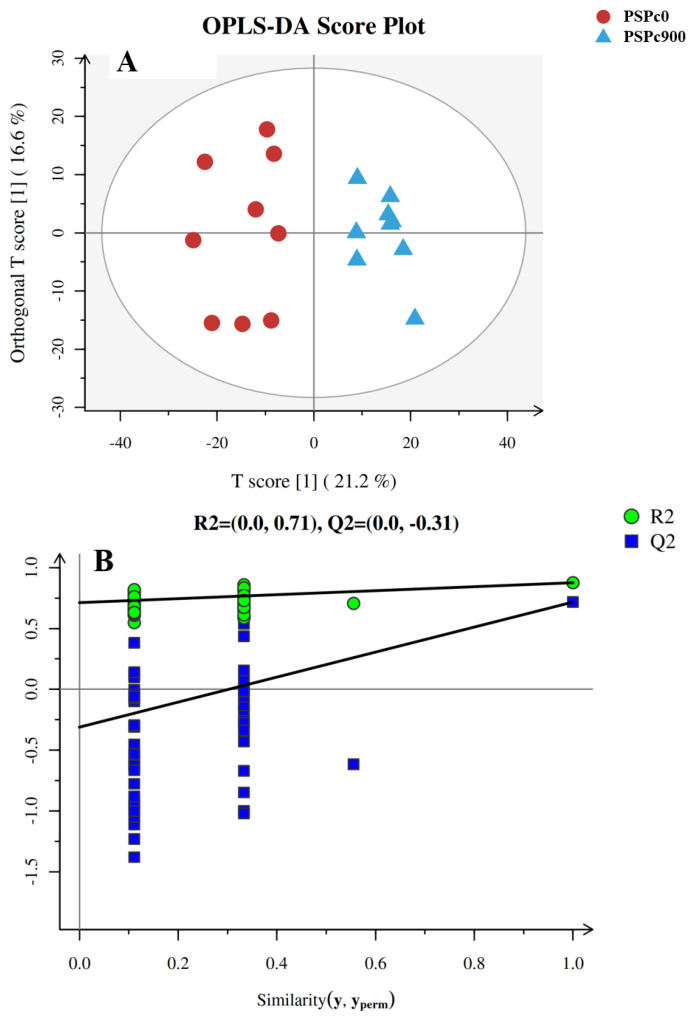
The score plots (**A**) and validation plots (**B**) of OPLS-DA from liver lipidomics profiles of juvenile American eels in the PSPc0 group and PSPc900 group. [1] in the image (**A**) means the first principal component.

**Figure 2 animals-12-02375-f002:**
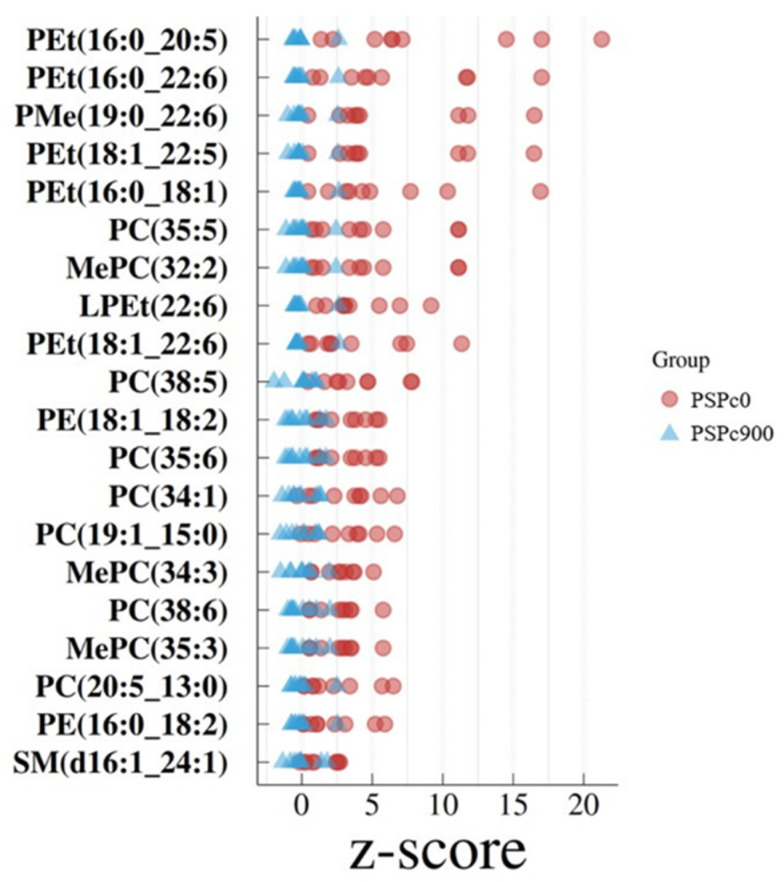
The Z-score plot of liver differential lipid metabolites of juvenile American eels in the PSPc0 group and PSPc900 group.

**Figure 3 animals-12-02375-f003:**
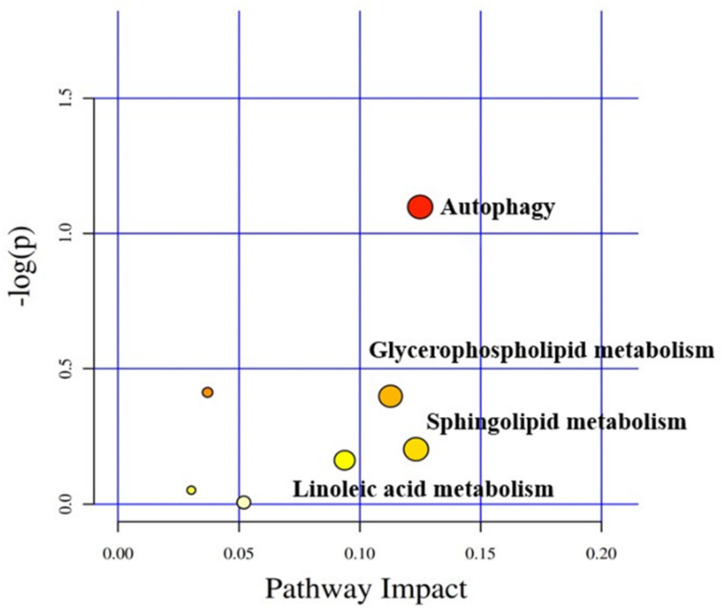
The pathway enrichment bubble plot of differential lipid metabolites in the liver of juvenile American eels in PSPc0 group and PSPc900 group. Each bubble in the figure is the expression of influence combined with the *p*-value. Different bubbles represent different metabolic pathways, and the larger the bubble, the greater the influence of the pathway. Different colors indicate different degrees of metabolic pathways, and the darker the color, the greater the changes in pathways. A higher −log(p) means a lower *p*-value. According to the importance and influence of metabolic pathways, the names of the most relevant metabolic pathways were labeled.

**Table 1 animals-12-02375-t001:** The growth performance of juvenile American eel in different PSPc groups.

Item	Groups				
	PSPc0	PSPc300	PSPc600	PSPc900	PSPc1200
IBW (g/fish)	10.48 ± 0.04 ^a^	10.50 ± 0.02 ^a^	10.54 ± 0.04 ^a^	10.45 ± 0.02 ^a^	10.53 ± 0.03 ^a^
FBW (g/fish)	16.40 ± 0.29 ^a^	17.45 ± 0.28 ^b^	17.96 ± 0.22 ^b^	19.22 ± 0.54 ^c^	17.97 ± 0.65 ^b^
WGR (%)	56.67 ± 2.45 ^a^	66.58 ± 3.28 ^b^	70.27 ± 1.89 ^b^	83.90 ± 5.44 ^c^	70.68 ± 5.67 ^b^
FE (%)	55.68 ± 1.89 ^a^	56.04 ± 2.44 ^a^	62.30 ± 2.90 ^b^	67.51 ± 3.14 ^c^	63.81 ± 2.23 ^bc^
FR (%)	1.41 ± 0.01 ^a^	1.51 ± 0.04 ^bc^	1.46 ± 0.05 ^ab^	1.56 ± 0.05 ^c^	1.44 ± 0.04 ^ab^
SR (%)	100 ^a^	96.67 ± 3.51 ^a^	99.00 ± 1.73 ^a^	100 ^a^	99.00 ± 1.73 ^a^

The values are means ± SD, *n* = 3. ^a,b,c^ Values with different superscripts in the same row indicate significant differences (*p* < 0.05). IBW, initial body weight; FBW, final body weight; WGR, weight gain rate; FE, feed efficiency; FR, feeding rate; SR, survival rate.

**Table 2 animals-12-02375-t002:** Proximate composition in whole fish of juvenile American eel in different PSPc groups.

Item	Groups				
	PSPc0	PSPc300	PSPc600	PSPc900	PSPc1200
Moisture (%)	69.03 ± 1.14 ^a^	68.37 ± 0.67 ^a^	69.13 ± 0.50 ^a^	68.20 ± 1.76 ^a^	68.27 ± 1.48 ^a^
Crude lipids (%)	9.94 ± 0.47 ^c^	9.64 ± 0.10 ^c^	9.00 ± 0.10 ^b^	8.41 ± 0.31 ^a^	8.82 ± 0.16 ^ab^
Crude protein (%)	18.44 ± 0.37 ^a^	18.78 ± 0.27 ^a^	18.85 ± 0.38 ^a^	18.88 ± 0.31 ^a^	18.69 ± 0.53 ^a^
Ash (%)	3.37 ± 0.26 ^a^	3.35 ± 0.07 ^a^	3.32 ± 0.20 ^a^	3.49 ± 0.19 ^a^	3.35 ± 0.17 ^a^

The values are means ± SD, *n* = 3. ^a,b,c^ Values with different superscripts in the same row indicate significant differences (*p* < 0.05).

**Table 3 animals-12-02375-t003:** Serum lipid parameters of juvenile American eel in different PSPc groups.

Item	Groups				
	PSPc0	PSPc300	PSPc600	PSPc900	PSPc1200
TC (mmol/L)	12.77 ± 1.46 ^b^	12.67 ± 0.40 ^b^	12.07 ± 0.45 ^ab^	10.67 ±0.33 ^a^	11.84 ± 1.39 ^ab^
TG (mmol/L))	7.13 ± 061 ^c^	5.90 ± 0.50 ^b^	5.04 ± 0.48 ^b^	3.77 ± 0.42 ^a^	5.21 ± 0.92 ^b^
LDL-C (mmol/L)	2.35 ± 0.30 ^c^	1.94 ± 0.12 ^b^	1.84 ± 0.24 ^b^	1.44 ± 0.22 ^a^	1.78 ± 0.11 ^ab^
HDL-C (mmol/L)	2.12 ± 0.36 ^a^	2.79 ± 0.13 ^b^	3.12 ± 0.60 ^b^	5.50 ± 0.33 ^c^	3.38 ± 0.23 ^b^

The values are means ± SD, *n* = 3. ^a,b,c^ Values with different superscripts in the same row indicate significant differences (*p* < 0.05). TC, total cholesterol; TG, triglyceride; HDL-C, high-density lipoprotein cholesterol; LDL-C, low-density lipoprotein cholesterol.

**Table 4 animals-12-02375-t004:** The activity or level of lipid metabolic enzymes in the liver of juvenile American eels in the different PSPc groups.

Item	Groups				
	PSPc0	PSPc300	PSPc600	PSPc900	PSPc1200
LPL (U/mgprot)	6.53 ± 0.31 ^a^	7.18 ± 0.17 ^a^	9.52 ± 0.33 ^b^	10.87 ± 0.77 ^c^	9.83 ± 0.52 ^b^
HL (U/mgprot)	9.06 ± 0.50 ^a^	9.18 ± 0.32 ^a^	11.42 ± 0.61 ^b^	13.00 ± 0.38 ^c^	11.24 ± 0.79 ^b^
TL (U/mgprot)	15.59 ± 0.22 ^a^	16.36 ± 0.47 ^a^	20.94 ± 0.89 ^b^	23.87 ± 1.15 ^c^	21.08 ± 1.09 ^b^
FAS (ng/mgprot)	1.76 ± 0.08 ^c^	1.27 ± 0.09 ^b^	1.20 ± 0.04 ^b^	1.05 ± 0.02 ^a^	1.21 ± 0.10 ^b^
ACC (ng/mgprot)	1.28 ± 0.10 ^c^	1.03 ± 0.10 ^b^	1.01 ± 0.12 ^b^	0.81 ± 0.09 ^a^	1.03 ± 0.04 ^b^

The values are means ± SD, *n* = 3. ^a,b,c^ Values with different superscripts in the same row indicate significant differences (*p* < 0.05). LPL, lipoprotein lipase; HL, hepatic lipase; TL, total lipase; FAS, fatty acid synthetase; ACC, acetyl-CoA carboxylase.

**Table 5 animals-12-02375-t005:** The differential lipid metabolites in the liver of juvenile American eels in the PSPc0 group and PSPc900 group.

Differential Lipid Metabolites	VIP	*p*-Value	log_2_ (FC)
Methyl phosphatidylcholine (MePC)			
35:3	12.7635	0.0092	0.3470
34:3	7.1823	0.0092	0.3913
32:2	4.9457	0.0092	0.6206
Phosphatidylcholine (PC)			
38:6	12.7635	0.0092	0.3470
19:1_15:0	12.1506	0.0092	0.2950
34:1	11.7509	0.0092	0.2760
38:5	7.2244	0.0092	0.3623
35:6	6.4893	0.0092	0.3596
35:5	4.9457	0.0092	0.6206
20:5_13:0	3.0010	0.0092	0.4176
Sphingomyelin (SM)			
d16:1_24:1	8.7552	0.0092	0.2740
Phosphatidylethanolamine (PE)			
18:1_18:2	6.4893	0.0092	0.3596
16:0_18:2	3.0517	0.0092	0.4240
Phosphatidylethanol (PEt)			
16:0_22:6	7.6820	0.0092	3.4540
18:1_22:6	5.5258	0.0092	3.1546
16:0_20:5	4.2147	0.0092	3.9162
16:0_18:1	3.5621	0.0092	3.3659
18:1_22:5	3.1935	0.0092	2.6640
Lysophosphatidylethanol (LPEt)			
22:6	4.3802	0.0092	3.0020
Phosphatidylmethanol (PMe)			
19:0_22:6	3.1865	0.0092	2.6425

VIP is the projection of the important value of variables in the OPLS-DA model with the threshold of 3; FC is the fold change, FC > 2: upregulation, FC < 0.67: downregulation.

**Table 6 animals-12-02375-t006:** The metabolic pathways of differential lipid metabolites in the liver of juvenile American eels in the PSPc0 group and PSPc900 group.

Pathway Name	Total	Hits	Lipid Metabolite	Up/Down
Autophagy	6	1	Phosphatidylethanolamine (PE)	Up
Sphingolipid metabolism	25	1	Ceramides (Cer)	Down
			Phosphatidylcholine (PC)	Up
Glycerophospholipid metabolism	52	3	Phosphatidylethanolamine (PE)	Up
			2-Lysophosphatidylcholine (2-LPC)	Down
Linoleic acid metabolism	28	1	Phosphatidylcholine (PC)	Up

Total, the number of lipids in the certain pathway; Hits, the lipids hit in the certain pathway; Up/down, upregulation/downregulation of lipid in PSPc900 group in comparison with PSPc0 group.

## Data Availability

All data generated for this study are included in the present article.

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
