# Peer review of "Effects of Dietary Supplementation of Peanut Skin Proanthocyanidins on Growth Performance and Lipid Metabolism of the Juvenile American Eel (Anguilla rostrata)"

_animals, 2022, doi:10.3390/ani12182375_

Round 1
Reviewer 1 Report
Peanut skin proanthocyanidins and grape seed proanthocyanidins have similar biological activities and can reduce the cost. In this paper, peanut skin proanthocyanidins were selected as feed additives to determine the effects of adding them on growth performance and lipid metabolism of juvenile American eel. In this study, dietary supplementation of an appropriate level of PSPc can promote the growth performance and regulate lipid metabolism of juvenile American eel, and dietary supplementation of 900 mg/kg PSPc has the best effect. The experimental design of this study is reasonable, and the results have certain guiding significance for the breeding of Juvenile American Eel.
Q1:There is no significant difference between PSPc600 group and PSPc1200 group in this paper. Have you considered setting a more precise gradient in the interval of PSPc 600-1200mg/kg again to verify the result that 900 mg/kg PSPc has the best effect?
Q2:In Table 1, why was the survival of PSPc300 and PSPc600 groups lower than that of PSPc0 group?
Q3:In Table 1, a and b at the initial body weight and the final body weight of the PSPc1200 group do not have superscripts.
Q4:Figure 1A and B is not very clear. It is recommended to replace the picture with a higher definition.
Q5:From the tenth line of Section 2.7 through Section 3.4,,the "P" representing a significant difference is not capitalized.
Q6:Determine whether the survival is expressed as "survival" or "survival rate".
Author Response
Q1:There is no significant difference between PSPc600 group and PSPc1200 group in this paper. Have you considered setting a more precise gradient in the interval of PSPc 600-1200mg/kg again to verify the result that 900 mg/kg PSPc has the best effect?
Re: many thanks for your advice. The more precise gradient in the interval of PSPc 600-1200mg/kg will be verified to confirm the result that 900 mg/kg PSPc might be the best in the future trial.
Q2:In Table 1, why was the survival of PSPc300 and PSPc600 groups lower than that of PSPc0 group?
Re: there is natural deaths in the long period trial. Although the values of survival rate of PSPc300 and PSPc600 groups were lower than that of PSPc0 group, they were similar with those the other groups. there was no significant difference in the survival rate among all the groups.
Q3:In Table 1, a and b at the initial body weight and the final body weight of the PSPc1200 group do not have superscripts.
Re: many thanks for your suggestion. We added the superscripts in the revised manuscript.
Q4:Figure 1A and B is not very clear. It is recommended to replace the picture with a higher definition.
Re: many thanks for your suggestion. We replaced these previous pictures with the new pictures with the higher definition and enlarged the picture. Please see the revised manuscript.
Q5:From the tenth line of Section 2.7 through Section 3.4, the "P" representing a significant difference is not capitalized.
Re: many thanks for your advices. We replaced those ‘P’ with italic ‘p’. they were identical in the revised manuscript.
Q6:Determine whether the survival is expressed as "survival" or "survival rate".
Re: it is expressed as "survival rate" in the revised manuscript.
Reviewer 2 Report
The manuscript entitled “Effects of Dietary Supplementation of Peanut Skin Proanthocyanidins on Growth Performance and Lipid Metabolism of the juvenile American Eel (Anguilla rostrata)”, authored by Yue Wang, Xue-hao Chen, Xin-yi Wu, Guo-he Cai, and Shao-wei Zhai, deals with the investigation of the effects derived from the dietary supplementation of peanut skin proanthocyanidins (PSPc) on growth parameters and lipid metabolism of juvenile American eel.
The manuscript is really well written, although many typos are present in the main text. Authors should very carefully reread the manuscript and fix as many as possible before the next submission. However, I do not think this can be a reason for not consider the manuscript suitable for publication.
Moreover, I would suggest a number of minor changes before consider the manuscript suitable for publication in Animals.
The abstract should be rewritten, paying close attention to the guidelines of the journal. In particular, this section should report a small state of the art, highlighting the relative problems or lacks currently present in this scientific field. From here, the authors should explain their motivation for producing this experimental trial. Finally, some numerical data related these experiments should be included, along with a sentencing conclusion at the end of this section.
The keyword section would also need to be rewritten. A maximum number of 10 words can be entered in this section. Word choice should fall on terms not contained in the title, and at most used in the abstract. The usefulness of keywords is to facilitate the search of the manuscript after publication using the most common scientific search motives. Consequently, I strongly suggest that authors eliminate words already used in the title, and replace them with other words that can make their article stand out after publication.
The introductory section is really too little detail.
(i) For example, the authors could better explain the chemistry of proanthocyanidins, which are very special polyphenolic compounds, almost unique in their structure (10.3390/antiox10081229; 10.1104/pp.20.00973).
(ii) The authors should better describe their use in fortified foods, perhaps not only in human nutrition. For example, the effect on lipid framework has been widely investigated in human diets (10.3390/antiox10081229; 10.1093/jn/nxz102).
(iii) Another point that the authors could better explain in the introduction relates to the distribution of this particular class of plant bioactive compounds in the plant kingdom, pointing out that not all plants are capable of producing them (10.3390/antiox10081229). Moreover, the major part of PACs is contained in the skins, seeds, or other waste products of the agribusiness industry. Consequently, the use of this waste product in the experiment conducted by the authors is very important from a circular economy point of view.
In 2.3. section, the authors should report equation according to the guidelines of the Journal. Moreover, each equation should be named with a number (e.g. (1), (2), etc...). Finally, authors should include the reference to the equation in the main text.
In section 2.5. the authors should better describe the experimental protocol used for the quantification.
In section 2.6. the authors should briefly report the chromatographic conditions and parameters used for the quantification. Moreover, how the quantification method, the use of standards, and other important information related to the quantification should be added in this paragraph.
The discussion section turns out to be too long. Strongly suggest that the authors try to divide this section into several sub-sections, assigning each one a title that briefly states the result.
Author Response
Many thanks for your confirmation and advices about improving the quality of our manuscript. We added the descriptions of chemistry, application, and distribution of proanthocyanidins according to your advices. Please see the revised manuscript.
In 2.3. section, the authors should report equation according to the guidelines of the Journal. Moreover, each equation should be named with a number (e.g. (1), (2), etc...). Finally, authors should include the reference to the equation in the main text.
Re: many thanks for your suggestion. In the revised manuscript, the equations were reported by equation insert function of Microsoft word according to the guidelines of the Journal, and the number and the references of those equations were also added.
In section 2.5. the authors should better describe the experimental protocol used for the quantification.
Re: many thanks for your suggestion. Some references were cited to describe the experimental protocol used in this section. Please see the revised manuscript.
In section 2.6. the authors should briefly report the chromatographic conditions and parameters used for the quantification. Moreover, how the quantification method, the use of standards, and other important information related to the quantification should be added in this paragraph.
Re: many thanks for your suggestion. The important information about the LC/MS for lipid metabolites quantification were added in the revised manuscript.
The discussion section turns out to be too long. Strongly suggest that the authors try to divide this section into several sub-sections, assigning each one a title that briefly states the result.
Re: many thanks for your suggestion. We divided this section into three sub-sections with the title.